# Intercalation Effects on the Dielectric Properties of PVDF/Ti_3_C_2_T_x_ MXene Nanocomposites

**DOI:** 10.3390/nano13081337

**Published:** 2023-04-11

**Authors:** Alexey Tsyganov, Maria Vikulova, Denis Artyukhov, Denis Zheleznov, Alexander Gorokhovsky, Nikolay Gorshkov

**Affiliations:** 1Department of Chemistry and Technology of Materials, Yuri Gagarin State Technical University of Saratov, 77 Polytecnicheskaya Street, 410054 Saratov, Russia; vikulovama@yandex.ru (M.V.); zheleznov_denis@internet.ru (D.Z.); algo54@mail.ru (A.G.); 2Department of Power and Electrical Engineering, Yuri Gagarin State Technical University of Saratov, 77 Polytecnicheskaya Street, 410054 Saratov, Russia; art@labmem.ru

**Keywords:** MXene, Ti_3_C_2_T_x_, high-k polymer nanocomposite, permittivity, dielectric loss, dielectric properties, polyvinylidene difluoride, conductive filler

## Abstract

In this study, we report the effect of intercalation of dimethyl sulfoxide (DMSO) and urea molecules into the interlayer space of Ti_3_C_2_T_x_ MXene on the dielectric properties of poly(vinylidene fluoride) (PVDF)/MXene polymer nanocomposites. MXenes were obtained by a simple hydrothermal method using Ti_3_AlC_2_ and a mixture of HCl and KF, and they were then intercalated with DMSO and urea molecules to improve the exfoliation of the layers. Then, nanocomposites based on a PVDF matrix loading of 5–30 wt.% MXene were fabricated by hot pressing. The powders and nanocomposites obtained were characterized by using XRD, FTIR, and SEM. The dielectric properties of the nanocomposites were studied by impedance spectroscopy in the frequency range of 10^2^–10^6^ Hz. As a result, the intercalation of MXene with urea molecules made it possible to increase the permittivity from 22 to 27 and to slightly decrease the dielectric loss tangent at a filler loading of 25 wt.% and a frequency of 1 kHz. The intercalation of MXene with DMSO molecules made it possible to achieve an increase in the permittivity up to 30 at a MXene loading of 25 wt.%, but the dielectric loss tangent was increased to 0.11. A discussion of the possible mechanisms of MXene intercalation influence on the dielectric properties of PVDF/Ti_3_C_2_T_x_ MXene nanocomposites is presented.

## 1. Introduction

Currently, polymer composites with high permittivity are attracting great attention due to the rapid development of electronics and their wide application potential in many electronic devices, such as capacitors for energy storage, communication devices, field-effect transistors, and actuators [1,2,3,4,5]. Such composites are usually made by loading ceramic or conductive fillers into dielectric polymer matrices, creating high-performance composites that combine the properties of flexibility and high permittivity [6].

Among the various polymer matrices, poly(vinylidene fluoride) (PVDF, chemical formula (C_2_H_2_F_2_)_n_) is the most popular research object because of its special dielectric properties, which permits its wide application in electrical devices [7]. PVDF has five different crystal forms (α, β, γ, δ, and ε phases) of which the β phase has the highest dipole moment and determines the permittivity value [8]. Although PVDF has high permittivity (ε′ = 11) compared to other polymers, it may be low for electronic applications. Therefore, significant efforts have been made to improve the dielectric properties of the PVDF polymer matrix. One of the most important approaches to increase permittivity is the loading of ceramic high-k particles, such as TiO_2_, titanates with perovskite structure, copper calcium titanate, and hollandite-like solid solutions, into the polymer matrix [9,10,11,12,13,14,15,16]. However, high permittivity of polymer/ceramic composites requires the loading of a high filler volume fraction (40 vol.% or more), which leads to an inevitable loss of flexibility and high dielectric losses. Thus, obtaining polymer composites with a high permittivity and a low filler volume fraction is a key problem in flexible dielectric materials. In order to achieve an increase in permittivity at a low filler volume fraction, various conductive fillers are loaded into polymer matrices. The most effective conductive fillers are carbon nanomaterials, such as carbon black, carbon nanotubes, graphene, and peelable graphene plates [17,18,19,20].

In recent years, in addition to carbon nanomaterials, a new group of two-dimensional materials, called MXenes, has demonstrated outstanding dielectric properties in polymer matrices. MXenes are layered carbides, carbonitrides, and nitrides of early transition metals with the general formula M_n+1_X_n_T_x_ (n = 1–3), where M is an early transition metal (such as Sc, Ti, Zr, Hf, V, Nb, Ta, Cr, Mo, and others), X is carbon and/or nitrogen, and T_x_ denotes the surface terminations (such as -OH, -O or -F) on the outer layer [21]. Due to their unique combination of electrical, mechanical, and other properties, as well as their extensive liquid-phase processing capabilities and controlled surface functionality, MXenes have generated great interest in various fields of science, especially energy storage, EMI protection, optoelectronics, water desalination, catalysis, medicine, and many others [22,23,24,25,26]. Due to their high electrical conductivity, MXenes have proven themselves as fillers for polymer matrices to effectively increase their permittivity [27,28,29]. Synthesis of MXenes is based on chemical selective etching of layers A from the layered hexagonal structures of M_n+1_AX_n_ phases (M is an early transition metal, A is an element of group IIIA or IVA, X is C and/or N, and n = 1–3) by treating them in HF solutions or other acidic solutions containing fluorine ions [30]. The morphology of MXenes is accordion-like stacks composed of close-packed layers. However, the morphology of MXenes can be changed by additional intercalation of their interlayer space with various ions and molecules [31]. The intercalation of MXenes results in an increase in the distance between the individual 2D sheets and, therefore, in an improvement in the further exfoliation of the 2D MXene sheets. In this case, the accordion-shaped stacks expand and spontaneously delaminate due to the weakening of the interaction force between the individual sheets. Homogenization of the filler in the bulk of the polymer matrix makes a significant contribution to the dielectric properties of the composites. Considering the complexity of filling the interlayer space of accordion-shaped MXenes with a polymer, the area of the inner-layer sheets can be made more accessible by intercalation.

The aim of this work is to study the effect of intercalation of Ti_3_C_2_T_x_ MXene with dimethyl sulfoxide (DMSO) and urea molecules on the dielectric properties of PVDF polymer matrix nanocomposites containing different amounts of Ti_3_C_2_T_x_ MXene fillers.

## 2. Materials and Methods

### 2.1. Synthesis of T_i3_AlC_2_ MAX Phase

Ti_3_AlC_2_ powder was synthesized by the molten salt method [32]. TiC (Mark B, TU 48-42-6-84, 0.8–1.5 µm, Mreda, Beijing, China), Ti (PTM-1, TU 14-22-57-92, Polema, Tula, Russia), and Al (PA 4, less than 100 µm, Nizhny Novgorod, Russia) powders with a molar ratio of 2TiC/Ti/Al was weighed and mechanically mixed in a Fritsch Pulverisette 6 mill using Si_3_N_4_ beaker and balls. Cylindrical granules with a diameter of 12 mm and a height of 20 mm were obtained from the crushed mixture of powders by uniaxial pressing at a pressure of 10 MPa. They were immersed in a NaCl–KCl eutectic melt preheated to 800 °C to create the conditions for blocking the supply of oxygen at early warm-up stages. At the same time, it was possible to place 2 granules into an allund crucible with a volume of 75 mL, which was completely filled with a salt melt. The filled crucible was covered with a lid to prevent the evaporation of the melt and kept in a muffle furnace at a temperature of 1300 °C for 3 h in an air atmosphere. After the reaction, the chamber was naturally cooled to room temperature. Then, the sample was sonicated several times in hot deionized water to remove hardened salts. As a result of these procedures, without additional grinding, a Ti_3_AlC_2_ MAX-phase powder was obtained.

### 2.2. Synthesis of Ti_3_C_2_T_x_ MXene

The 2D Ti_3_CT_x_ MXene powder was obtained by hydrothermal-assisted etching processing in accordance with the procedure described in previous works [33,34]. This method was chosen because hydrothermal conditions have many advantages, such as providing a closed reaction environment with a high temperature and a high pressure. At high temperature and pressure, fluoride salts can be better dissolved in water, thereby promoting the etching reaction and decreasing the loss of etching solutions. Under the conditions of high temperature and high pressure, the solvent in the solution is in a critical or supercritical state, and the reaction activity increases. Potassium fluoride (KF∙2H_2_O, 99.5, UniChim, Saint-Petersburg, Russia) and hydrochloric acid (HCl, 11.8 mol/l, NizhHimProm, Nizhny Novgorod, Russia) were used as the raw materials for etching the Ti_3_AlC_2_ MAX phase. First, 11 g of potassium fluoride and 60 mL of deionized water were added to 60 mL of concentrated hydrochloric acid. After that, 6 g of Ti_3_AlC_2_ was immersed in the prepared etching solution in a 400 mL Teflon autoclave. The autoclave was sealed and kept at a temperature of 150 °C for 20 h, and then cooled to room temperature naturally. Then, the mixture was washed with deionized water and filtered several times until the pH = 6. The resulting powder was dried at 80 °C for 12 h. As a result of these procedures, Ti_3_C_2_T_x_ MXene powder was obtained.

### 2.3. Intercalation of Ti3C2Tx MXene

The following procedure was used to intercalate the organic molecules, DMSO (99.5% (CH_3_)_2_SO, dimethylsulfoxide, TU 2635-114-44493179-08, VitaHim, Dzerzhinsk, Russia) and urea ((NH_2_)_2_CO, Mark B, UralChem branch, Moscow, Russia), into the interlayer space of MXene: 2 g of Ti_3_C_2_T_x_ MXene powder was mixed with 50 mL of DMSO or 50 mL of 20% aqueous solution of urea, which was then stirred on a magnetic stirrer for 48 h at room temperature. Later, the resulting colloidal solutions were washed with deionized water, filtered, and dried at 80 °C. The resulting samples are hereinafter referred to as Ti_3_C_2_T_x_ (DMSO) and Ti_3_C_2_T_x_ (urea).

### 2.4. PVDF/MXene Nanocomposites Preparation

Poly(vinylidene fluoride) (PVDF, Arkema, Kynar 761, Colombes, France) was used as the polymer matrix for nanocomposite preparation. N, N-Dimethylformamide (DMF, C_3_H_7_NO, analytical grade, Ekos-1, Moscow, Russia) was used as the PVDF solvent.

For the PVDF/MXene nanocomposite creation, 5 wt.% solution of PVDF in DMF was prepared, and then Ti_3_C_2_T_x_, Ti_3_C_2_T_x_ (DMSO), and Ti_3_C_2_T_x_ (urea) powders were added. The conductive filler content was from 5 to 30 vol.%., and the ultrasonic treatment was performed for 1 h to improve the dispersion of the filler in the PVDF matrix. Then, the PVDF solution with the filler was left stirring at 60 °C for 24 h. The resulting dispersion was poured into deionized water to remove the solvent. After removing water with the solvent, the sponge-like polymer substance was dried at 120 °C for 8 h. The nanocomposite samples were fabricated in the form of disks with 12 mm in diameter and ~0.7 mm in thickness by uniaxial hot pressing at a temperature of 180 °C and a pressure of 10 MPa for 1 h.

### 2.5. Characteristic Methods

The ARL X’TRA device (Thermo Scientific, Ecublens, Switzerland) using Cu Kα radiation (λ = 0.15412 nm) was used to record the diffractogram of the MAX phase and MXenes. The morphology of the MXenes, as well as the distribution of the fillers and the structural features of the polymer matrix composites, was investigated using an ASPEX Explorer scanning electron microscope (ASPEX, Framingham, MA, USA). Fourier-transform infrared spectroscopy (FTIR) was carried out using an FT-801 FTIR spectrometer (Simex, Novosibirsk, Russia).

The dielectric properties of the nanocomposites were studied by impedance spectroscopy using Novocontrol Alpha AN Impedance Analyzer (Novocontrol Technologies GmbH & Co. KG, Montabaur, Germany). The experimental values of the real and imaginary parts of the impedance (Z′ and Z″) were obtained, from which the values of permittivity (ε′, ε″ and ε), dielectric loss (tanδ), and conductivity were calculated (σ′, σ″ and σ). The reproducibility of the results obtained was ensured by successive measurements of three independent samples of each composition, while the relative error did not exceed 5%. The measurements were carried out in the frequency range from 1 to 10^6^ Hz, and the voltage amplitude was 100 mV. A silver-containing paste (trademark Contactol K13, OOO Adecvat, Moscow, Russia) was applied to the surface of the nanocomposite samples as electrodes, followed by drying at 120 °C.

## 3. Results and Discussion

The X-ray diffraction pattern of the titanium–aluminum carbide powder obtained in the eutectic melt of NaCl-KCl salts at a temperature of 1300 °C for 3 h is presented in Figure 1. It is seen that the main product of the resulting powder is the Ti_3_AlC_2_ MAX phase (JCPDS 52-0875); however, peaks are also observed on the diffraction pattern, confirming the presence of an impurity Ti_2_AlC phase in a small amount. The X-ray diffraction pattern of the T_3_C_2_T_x_ MXene obtained after the hydrothermal treatment of the Ti_3_AlC_2_ MAX phase in a reaction system from the solution containing a mixture of KF and HCl is also presented. After the hydrothermal treatment, all diffraction peaks of the MAX phase disappear, and one broad peak appears at 2θ = 7.78°, which can be attributed to the reflection from the (002) plane of the T_3_C_2_T_x_ MXene layered structure. For the initial Ti_3_AlC_2_ MAX phase, the reflection angle (002) is 2θ = 9.4°; thus, it can be concluded that the lattice parameter *c* has increased due to the destruction of the Ti-Al bond and the etching of the Al atomic layer from the structure of the Ti_3_AlC_2_ MAX phase. It is known that an increase in the d-spacing of MXene further improves the exfoliation of 2D MXene sheets, which is achieved in this case by intercalation with DMSO and urea molecules. Upon intercalation of DMSO and urea, the (002) peaks shift to smaller angles of up to 2θ = 6.32° in both cases, which corresponds to an increase in d-spacing. In addition, the peaks become more intense due to the greater thickness of the MXene layers.

The SEM images of the initial Ti_3_C_2_T_x_ MXene (Figure 2a,b) obtained after etching of the Ti_3_AlC_2_ MAX phase and the samples intercalated with DMSO (Figure 2c) and urea molecules (Figure 2d) are presented in Figure 2. As can be seen (Figure 2a,b), for the resulting Ti_3_C_2_T_x_, an accordion-like morphology is observed. The 2D layers give densely packed particles, which confirm the successful removal of Al layers from the Ti_3_AlC_2_ structure. The intercalation of Ti_3_C_2_T_x_ in both cases (Figure 2c,d) leads to a markedly improved degree of exfoliation and larger gaps between the sheets. In addition, it can be observed that intercalation leads to the appearance of thick multilayers formed as a result of the bonding of individual monolayers. After the appearance of a certain number of thick layers, which spontaneously delaminate due to the weakening of the interaction force, the accordion-shaped particles of Ti_3_C_2_T_x_ expand.

The FTIR spectra corresponding to the pure PVDF, Ti_3_C_2_T_x_ MXene, and polymer nanocomposites with different MXene loadings are shown in Figure 3. As can be seen, clearly pronounced peaks cannot be distinguished in the FTIR spectrum of the Ti_3_C_2_T_x_ MXene powder. At the same time, the FTIR spectra of the pure PVDF and PVDF/Ti_3_C_2_T_x_ composites with different loadings of Ti_3_C_2_T_x_ show peaks confirming the formation of a mixture of PVDF polymorphs. The main peaks of the absorption bands of the PVDF polymorphs are observed in the range from 750 to 1410 cm^−1^, which can be used to identify the presence of the α-phase (763, 883, 1073 и 1410 cm^−1^), β-phase (840, 883, 1073, 1273 и 1410 cm^−1^), and γ-phase (883, 1073 и 1410 cm^−1^) [35,36]. It should be noted that in the manufacture of composites based on PVDF, it is important to ensure the predominance of the ferroelectric β-phase, in contrast to the non-polar α-phase. In the FTIR spectra of the PVDF/Ti_3_C_2_T_x_ nanocomposites with different filler loadings, intense peaks of α- and β-phases are observed. Therefore, it can be noted that the introduction of the Ti_3_C_2_T_x_ MXene filler into the PVDF matrix does not change the phase composition of PVDF.

The SEM images of the cross sections of the PVDF/Ti_3_C_2_T_x_, PVDF/Ti_3_C_2_T_x_ (DMSO), PVDF/Ti_3_C_2_T_x_ (urea) nanocomposites at the filler loading of 20 wt.% are shown in Figure 4. As can be seen, in all cases, the MXene fillers are accordion-shaped stacks and lamellae, resulting from the transverse shear of the multi-layer MXene. In addition, agglomerated particles are not observed in the cross sections of the composites, and the filler particles are evenly distributed over the volume of the polymer matrix. As expected, MXene particles have good compatibility with the PVDF polymer matrix, which can be explained by the abundance of functional groups (-F, -OH, and -O) on the surface of the Ti_3_C_2_T_x_ layers, which contribute to high adhesion due to the possible formation of chemical bonds between Ti_3_C_2_T_x_ particles and the PVDF matrix. For the PVDF/Ti_3_C_2_T_x_ (DMSO) sample, it can also be seen that the polymer phase can penetrate into the space between the layers formed as a result of exfoliation. This may be because PVDF dissolves well in DMSO during the preparation of the composites.

The permittivity and dielectric loss tangent of the nanocomposites with the initial filler PVDF/T_3_C_2_T_x_ and MXenes, intercalated with the organic molecules PVDF/T_3_C_2_T_x_(DMSO) and PVDF/T_3_C_2_T_x_(urea) depending on the filler loading are shown in Figure 5.

As can be seen for all types of nanocomposites, as expected, there is an increase in the permittivity and dielectric loss with an increase in the filler loading. When the initial loading of Ti_3_C_2_T_x_ MXene into the PVDF matrix is up to 25 wt.%, the permittivity increases uniformly from 8 to 24, with a moderate increase in dielectric loss from 0.016 to 0.033 at a frequency of 1 kHz. A further increase in the loading of Ti_3_C_2_T_x_ to 30 wt.% leads to a sharp increase in the permittivity to 37, but the dielectric losses also increase sharply to 0.093. For the potential application of high-k composites in electronic devices, high permittivity and low dielectric losses must be ensured. Therefore, in this work, with a sharp increase in dielectric losses, further loading of the fillers was not applied. This behavior in the growth of the permittivity of polymer composites with a conductive filler is attributed to the formation of a network of nanocapacitors, in which neighboring fillers act as electrodes, and the polymer matrix acts as a dielectric layer [37,38]. Therefore, at first, when a small amount (5 wt.%) of T_3_C_2_T_x_ MXene is dispersed in the PVDF matrix, the formation of nanocapacitor structures does not occur due to the large distance between MXene particles. With an increase in the loading of the conductive filler, the distance between the electrodes is decreased and a network of nanocapacitors uniformly distributed over the composite volume is formed. It leads to an increase in the intensity of the electric field around the MXene flakes and polarizes the PVDF matrix as a dielectric layer. The second criterion for such an increase in the permittivity is an increase in the Maxwell–Wagner–Sillars interfacial polarization [39], which occurs due to a large difference in electrical conductivities between the MXene conductive filler and the PVDF-insulating polymer matrix.

The use of intercalated MXenes in the composites leads to significant changes in the permittivity and dielectric loss tangent, which are more clearly demonstrated in Figure 6.

The use of MXenes intercalated with urea molecules makes it possible to increase the permittivity and decrease dielectric losses, especially at a high filler loading. Specifically, at loadings of 25 and 30 wt.%, intercalation with urea leads to an increase in the permittivity from ε′ = 22 to ε′ = 27 and from ε′ = 38 to ε′ = 42, respectively. At the same time, at loadings of 20 and 25 wt.%, a decrease in dielectric losses is observed with a value of 0.02 and 0.03, respectively. In the case of intercalation of MXenes with DMSO molecules, a sharp increase in the permittivity at a filler loading of 25 wt.% can be observed; however, in this case, a sharp increase in dielectric losses is also observed, which has an unacceptable value of tgδ = 0.11 at a loading of 25 wt.%.

It should be noted that the frequency dependences of the permittivity and dielectric losses of the PVDF/MXene nanocomposites are similar to the dependence of the pure PVDF. The permittivity at a moderate filler loading (up to 30 wt.% Ti_3_C_2_T_x_ and Ti_3_C_2_T_x_ (urea), and up to 25 wt.% Ti_3_C_2_T_x_ (DMSO)) weakly depends on the frequency in the range of 10^2^–10^5^ Hz, and it shows a slight decrease with a further increase in frequency. At frequencies below 100 Hz, the permittivity shows a small increase due to the effect of electrode polarization. In this case, the dielectric loss tangent has a minimum value at frequencies of 10^3^–10^5^ Hz, and then it shows a tendency toward a sharp increase. This is due to the fact that the polarization rate of a large number of PVDF dipoles cannot keep up with changes in the external electric field, which leads to relaxation losses and an increase in dielectric losses at high frequencies. As can be seen, at filler concentrations above 25 wt.%, a clear dependence of the permittivity on frequency is observed in the entire range under study. This can be attributed to the Maxwell–Wagner–Sillars interfacial polarization as a result of the accumulation of charge carriers at the MXene/PVDF interfaces [18,40,41]. It is known that dielectric losses can be mainly caused by conduction losses, relaxation polarization losses during space charge motion in the low-frequency range, and dipole motion in the high-frequency range, as well as resonant losses under the action of an applied electric field. In this case, the increase in dielectric losses with an increase in the concentration of MXene in the nanocomposites is associated primarily with the formation of electrically conductive tracks, which, due to the tunneling effect, can form even though MXene particles are not connected to each other in the polymer matrix [18,42].

The frequency dependences of conductivity for the PVDF/MXene nanocomposites are shown in Figure 7. As can be seen, the measured conductivity tends to increase with increasing frequency. At a frequency of 1 Hz, the conductivities do not exceed 10^−10^ S·cm^−1^, and as the frequency increases to 1 MHz, the conductivities increase to 10^−6^–10^−5^ S·cm^−1^. In addition, along with the dielectric losses, the conductivity exhibits an increase with increasing concentration of the conductive filler due to the formation of electrically conductive paths.

An increase in the permittivity with the use of intercalated T_3_C_2_T_x_ MXenes can be associated with improved dispersion and an increase in the surface area of nanoelectrodes (MXene flakes) formed by the network of nanocapacitors. Since hydrogen bonds between the MXene layers greatly complicate their exfoliation and prevent the penetration of the polymer phase into close-packed accordion-like particles, the dielectric properties of such nanocomposites are somewhat limited. The intercalation of DMSO and urea molecules into the interlayer space of MXenes weakens the interlayer interactions and makes it possible to change their accordion-like morphology to thicker multilayer lamellae with fewer gaps between the stacks. Furthermore, the PVDF polymer matrix can easily penetrate into the large gaps of the MXenes while providing a large contact area between the PVDF matrix and the MXene flakes. The network of nanocapacitors built in this way provides a larger capacitance for storing electric charge and has a higher permittivity. In the case of the MXene intercalated with DMSO molecules, the increase in the permittivity and the increase in the dielectric loss can be attributed to several factors. Firstly, DMSO intercalation can promote slip-induced rupture of some MXene layers during hot pressing, resulting in the formation of conductive paths in the PVDF/Ti_3_C_2_T_x_ (DMSO) composites and, consequently, an increase in dielectric losses associated with charge leakage. Secondly, as a result of the interaction of Ti_3_C_2_T_x_ (DMSO) with the PVDF matrix, deintercalation of DMSO molecules from the MXene interlayer space can occur. This is because DMSO is a good solvent for PVDF. Thus, during hot pressing, some of the DMSO molecules pass from the MXene interlayer space to the PVDF polymer matrix. In this case, further removal of the DMSO solvent from the composites can lead to the appearance of a large number of pores and defects at the MXene/PVDF interface, which leads to an increase in the dielectric losses of the PVDF/Ti_3_C_2_T_x_ (DMSO) composites. Therefore, the maximum allowable concentration of the filler Ti_3_C_2_T_x_ (DMSO) is limited to 20 wt.%. This is due to the fact that a further increase in the Ti_3_C_2_T_x_ (DMSO) filler leads to a sharp and undesirable increase in dielectric losses. In addition, the Ti_3_C_2_T_x_ and Ti_3_C_2_T_x_ (urea) fillers show higher efficiency in improving dielectric properties at all filler concentrations. Hence, we conclude that the use of MXenes intercalated with DMSO molecules is not advisable in the fabrication of composites based on a PVDF matrix.

With a melting point of around 180 °C and extrusion properties, PVDF is widely used as a 3D printing filament. Thus, PVDF/MXene-based nanocomposites with a high permittivity and a low loss tangent can be used in the development of high-k components for flexible electronics using additive 3D printing technologies.

## 4. Conclusions

In this study, the Ti_3_AlC_2_ MAX phase was successfully synthesized in a KCl-NaCl eutectic melt at a relatively low temperature of 1300 °C under an air atmosphere. Using a simple hydrothermal treatment of Ti_3_AlC_2_ powder in a mixture of KF and HCl, Al atomic layers were successfully removed from the structure of the MAX phase and, as a result, Ti_3_C_2_T_x_ MXene was obtained, which was subsequently intercalated with DMSO and urea molecules. It was shown by the SEM and XRD methods that the obtained MXene had an accordion-like morphology, in which 2D layers formed densely packed particles. At the same time, further intercalation of the MXene led to an increase in d-spacing and an improvement in the exfoliation of the 2D layers; as a result, thick multilayer particles appeared, which spontaneously delaminated due to the weakening of hydrogen bonds between the layers. Nanocomposites, containing 5 to 30 wt.% of the conductive filler, were fabricated by the hot-pressing method of MXenes in the PVDF polymer matrix. It was shown by the SEM method that the studied MXenes had good compatibility with the PVDF matrix and were uniformly dispersed in it. Using impedance spectroscopy at frequencies of 10^2^–10^6^ Hz, it was shown that the nanocomposites with a loading of 25 wt.% initial Ti_3_C_2_T_x_ had an increase in the permittivity up to ε′ = 22, with a small dielectric loss tangent of tan δ = 0.033. Additional intercalation of MXene with urea molecules made it possible to increase the permittivity to ε′ = 27 at a loss tangent of tan δ = 0.030. Intercalation of MXene with DMSO molecules, on the contrary, led to an increase in the permittivity up to ε′ = 32; however, the dielectric losses had a high value of tan δ = 0.11. The explanation for the improvement in the dielectric property was based on the accumulation of charges at the interface between the MXene flakes and the PVDF matrix, as well as the formation of a network of nanocapacitors in which the MXene flakes were electrodes separated by a polymer dielectric. The effect of intercalation on the dielectric properties of the nanocomposites was associated with an increase in the surface area of MXenes and an improvement in the exfoliation of their layers when dispersed in a PVDF matrix. In this case, the inevitable increase in dielectric losses with an increase in the filler loading was associated with the formation of electrically conductive tracks, which, due to the tunneling effect, could form even though the MXene particles were not connected to each other in the polymer matrix. The materials studied in this work, as well as the methods for their production, can be used in the development of high-k flexible electronic components using additive 3D printing technologies.

## Figures and Tables

**Figure 1 nanomaterials-13-01337-f001:**
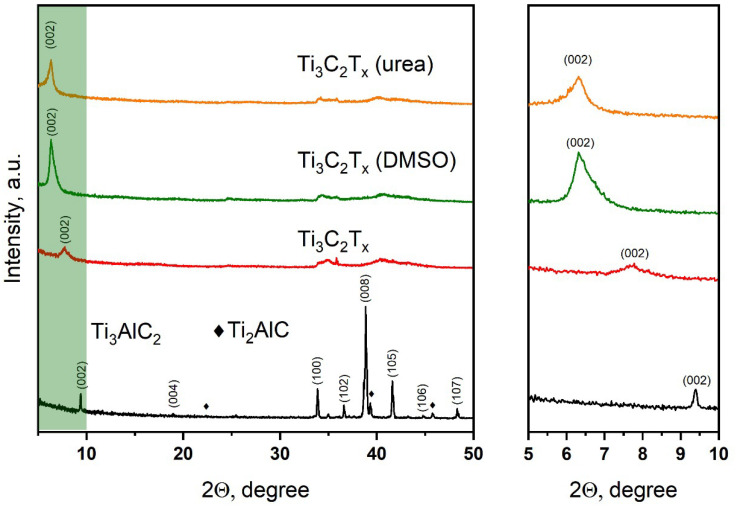
XRD patterns of MAX powder (Ti_3_AlC_2_), MXene powder (Ti_3_C_2_T_x_), and Ti_3_C_2_T_x_ MXene intercalated with DMSO and urea.

**Figure 2 nanomaterials-13-01337-f002:**
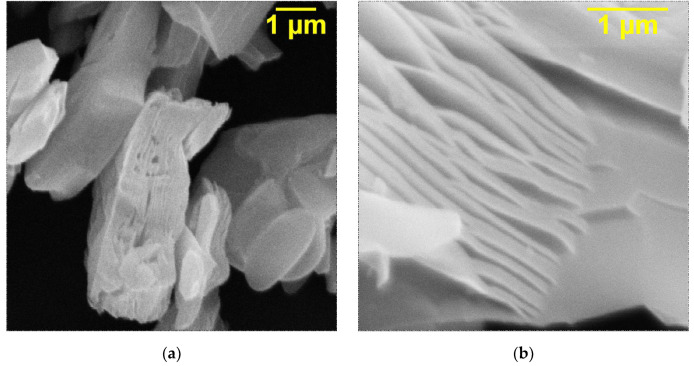
SEM images of (**a**,**b**) Ti_3_C_2_T_x_ MXene and Ti_3_C_2_T_x_ MXene intercalated with (**c**) DMSO and (**d**) urea molecules.

**Figure 3 nanomaterials-13-01337-f003:**
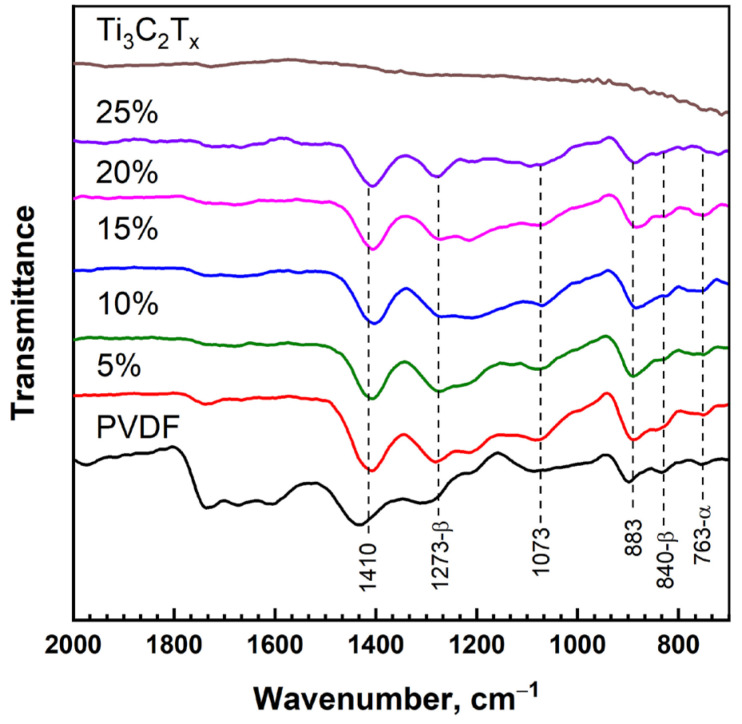
FTIR spectra of pure PVDF, Ti_3_C_2_T_x_ MXene, and polymer nanocomposites with different MXene loadings.

**Figure 4 nanomaterials-13-01337-f004:**
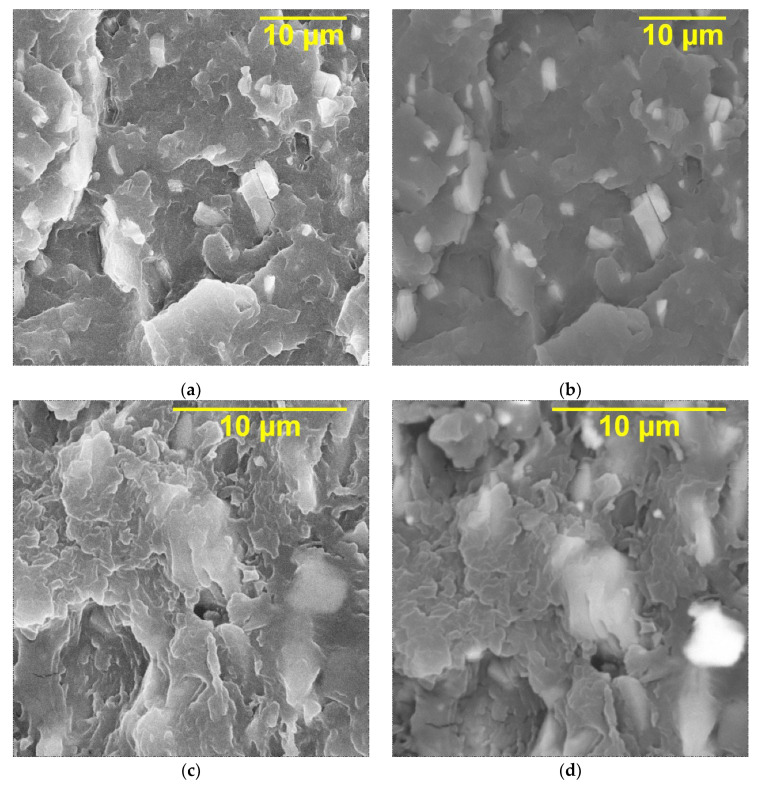
SEM images using primary electrons (left) and secondary electrons (right) of cross sections of (**a**,**b**) PVDF/Ti_3_C_2_T_x_, (**c**,**d**) PVDF/Ti_3_C_2_T_x_ (DMSO), and (**e**,**f**) PVDF/Ti_3_C_2_T_x_ (urea) with a filler loading of 20 wt.%.

**Figure 5 nanomaterials-13-01337-f005:**
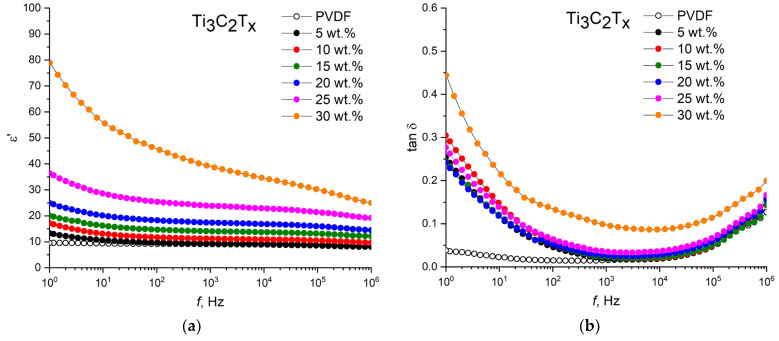
Frequency dependences of permittivity of (**a**) PVDF/Ti_3_C_2_T_x_, (**c**) PVDF/Ti_3_C_2_T_x_ (DMSO), and (**e**) PVDF/Ti_3_C_2_T_x_ (urea) and dielectric loss of (**b**) PVDF/Ti_3_C_2_T_x_, (**d**) PVDF/Ti_3_C_2_T_x_ (DMSO), and (**f**) PVDF/Ti_3_C_2_T_x_ (urea).

**Figure 6 nanomaterials-13-01337-f006:**
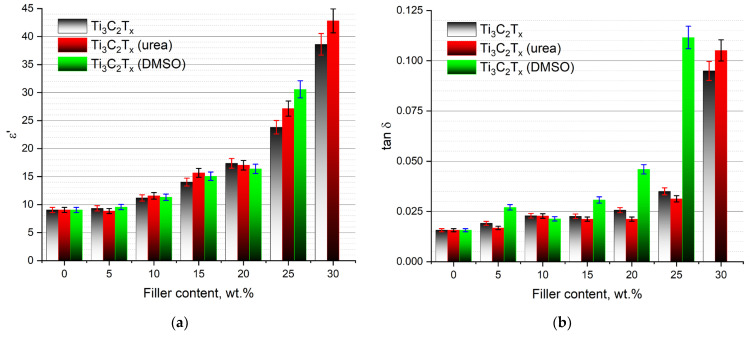
The dependences of (**a**) the permittivity and (**b**) dielectric losses of the PVDF/MXene nanocomposites on the filler loading at a frequency of 1 kHz.

**Figure 7 nanomaterials-13-01337-f007:**
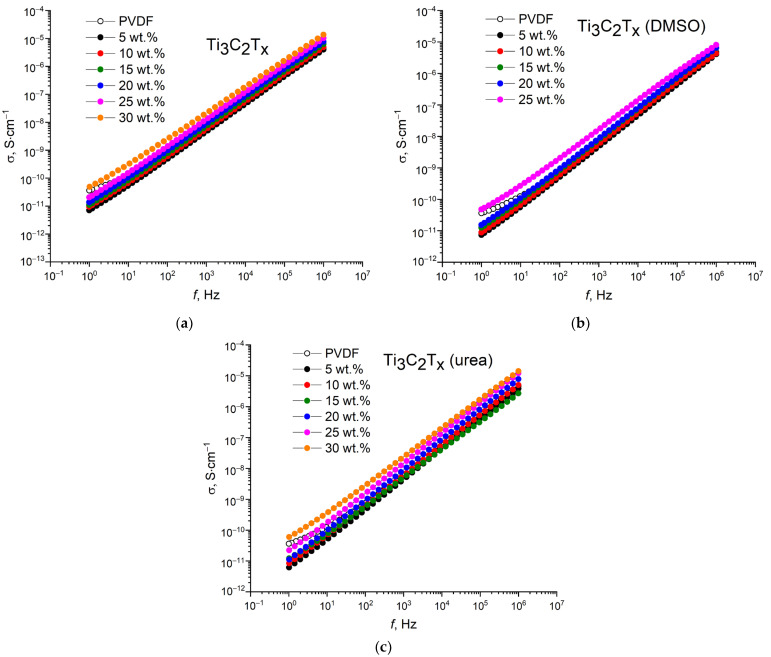
Frequency dependences of conductivity of (**a**) PVDF/Ti_3_C_2_T_x_, (**b**) PVDF/Ti_3_C_2_T_x_ (DMSO), and (**c**) PVDF/Ti_3_C_2_T_x_ (urea) nanocomposites.

## Data Availability

Not applicable.

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
