# Peer review of "Intercalation Effects on the Dielectric Properties of PVDF/Ti3C2Tx MXene Nanocomposites"

_nanomaterials, 2023, doi:10.3390/nano13081337_

Round 1

Reviewer 1 Report

The authors studied preparation and characterization of interesting nanocomposites based on PVDF and MXene. The products were firstly characterized by means of X-ray diffraction, infrared and scanning electron microscopy techniques. Then, the dielectric properties, expressed by permittivity and dielectric loss parameters, were investigated. The manuscript is well written and clear. The paper may be published in Nanomaterials, however, after some important corrections, as justified in the following points:

1. The permittivity and dielectric loss parameters were recorded in a frequency window between 100 Hz and 1 MHz. However, at lower frequencies, some important information may be revealed. Please show the ε’ and tanδ at frequencies between 1 Hz and 1 MHz.

 2. The conductivity is an important aspect that should be taken into consideration for practical uses, such as electronic devices. Therefore, I consider that the frequency-dependent conductivity spectra need to be included. It is important to observe the ac and dc components of the conductivity, and, eventually, the MWS polarization. Some of relevant articles may be considered, such as: Polymer 149, 73-84, 2018 (doi.org/10.1016/j.polymer.2018.06.061);  Polymer 203, 122785, 2020 (doi.org/10.1016/j.polymer.2020.122785).

 3. The authors did not study any thermal properties for the products (e.g. thermogravimetric (TG) and differential scanning calorimetry (DSC) measurements); the intercalation effects on thermal properties of final products should be interesting and may enhance the quality of the manuscript.

Reviewer 3 Report

This article is comprehensive, logically organized, and contains valuable information on the intercalation effects on the dielectric properties of PVDF/Ti3C2Tx MXene nanocomposites. The authors presented the dependences of (a) the permittivity and (b) dielectric losses of PVDF/MXenes nano-composites on the filler loading at a frequency of 1 kHz in Figure 6. It is suggested that the authors should provide the error bars of the permittivity and dielectric losses of PVDF/MXenes nano-composites so that the readers will have an idea of the reproducibility of the data. The submitted manuscript has significant scientific insights and the experimental data support the conclusions. However, the present submission requires minor revisions before being considered for publication in the esteemed Nanomaterials in its current condition. I hope the authors will find my comments helpful.

Round 2

Reviewer 1 Report

I agree with the revision version. In my opinion, the manuscript can now be published in Nanomaterials.